# High-energy electron emission from metallic nano-tips driven by intense single-cycle terahertz pulses

Sha Li[1] & R.R. Jones[1]

Electrons ejected from atoms and subsequently driven to high energies in strong laser fields enable techniques from attosecond pulse generation to imaging with rescattered electrons. Analogous processes govern strong-field electron emission from nanostructures, where long wavelength radiation and large local field enhancements hold the promise for producing electrons with substantially higher energies, allowing for higher resolution time-resolved imaging. Here we report on the use of single-cycle terahertz pulses to drive electron emission from unbiased nano-tips. Energies exceeding 5 keV are observed, substantially greater than previously attained at higher drive frequencies. Despite large differences in the magnitude of the respective local fields, we find that the maximum electron energies are only weakly dependent on the tip radius, for 10 nm $< R <$ 1,000 nm. Due to the single-cycle nature of the field, the high-energy electron emission is predicted to be confined to a single burst, potentially enabling a variety of applications.

[1] Department of Physics, University of Virginia, 382 McCormick Road, Charlottesville, Virginia 22904, USA. Correspondence and requests for materials should be addressed to S.L. (email: sl9es@virginia.edu) or to R.R.J. (email: bjones@virginia.edu).

Optical field-driven electrons are a hallmark of intense laser physics and attosecond science[1,2]. They are the basis for attosecond pulse generation and enable a variety of techniques for probing dynamics with atomic-scale temporal and/or spatial resolution[3–11]. However, for condensed matter targets, laser-induced surface and bulk damage limit the maximum visible or near-infrared drive intensities that can be applied, reducing the magnitude of the electronic momentum transfer and the viability of these probe methods. To combat this issue, enhanced local fields in the vicinity of metallic nanostructures and longer wavelength drivers can be exploited to significantly reduce the optical intensities required to induce emission and accelerate electrons to high energies. Indeed, during the last decade, numerous studies have explored strong-field physics by coupling femtosecond laser pulses to nanostructured solids, particularly metallic nano-tips[12–26]. The results of those investigations demonstrated that, with proper modification, much of the basic physics underlying laser-induced ionization of atomic and molecular gases also applies to laser-induced electron emission from solids. Specifically, different mechanisms, including photoemission, photo-field emission and optical field emission, dominate in different frequency regimes, and the well-known Keldysh parameter[27] can be adapted to define the transition from photon- to field-induced processes[12]. The classical notion of field-driven electron dynamics, originally employed to understand the response of gas-phase systems to intense laser pulses, can be readily applied to nanostructured targets as well. Examples of related phenomena include ponderomotive acceleration and electron rescattering of ejected electrons[15,25], carrier-envelope phase effects[15,25] and extreme ultraviolet fluorescence induced by strong-field ionization[28–30]. That said, the substantial local field enhancements in the vicinity of nanostructured materials can lead to key differences in the details of how, and how much, energy and momentum are transferred to the driven electrons[12,22,24,25]. For example, when exploring electron emission from gold nano-tips as a function of drive laser wavelength, Herink et al.[12] observed a transition from the familiar quiver response of electrons in an oscillating field, to a new sub-cycle energy transfer regime. At sufficiently long wavelengths, ejected electrons leave the vicinity of the nano-tip (and the enhanced field) in a fraction of the optical cycle, resulting in a maximum energy transfer that is proportional to the drive field at the instant of emission.

At still longer wavelengths, terahertz radiation has recently been employed to control photoemission from nano-tips[31] and as a temporal gate to enable time-resolved scanning tunnelling microscopy[32]. While microwave-induced field emission has been well studied[33,34], electron field emission in the presence of terahertz radiation alone has only very recently been reported, due primarily to the lack of sufficiently intense table-top terahertz sources[35,36]. Further investigation of terahertz-induced emission and subsequent energy transfer to, and beyond, the keV scale is motivated by fundamental science as well as applications, including time-resolved atomic-scale imaging where the smallest resolvable spatial features scale as the inverse of the electron momentum. Electrons driven to high energies in the enhanced local field of metalic nano-tips can provide spatially and temporally localized electron bursts of high-energy electrons for directly probing matter[35,37]. In contrast to the optical or infrared regimes, at terahertz frequencies true single-cycle drive fields can be utilized, confining the high-energy electron emission to a single, isolated burst.

From a fundamental science perspective, terahertz photon energies are three orders of magnitude smaller than the work functions of typical metals, so ultra-fast emission at terahertz frequencies enables access to purely field-driven, rather than photon-mediated processes. In metals with sufficiently low scattering, coherent energy transfer to electrons prior to ejection could result in highly non-equilibrium conditions which, among other things, might lead to substantial deviations of the emission process from purely adiabatic tunnelling[36]. Moreover, at terahertz frequencies, the local electron-field interaction can be pushed far deeper into the sub-cycle regime as compared with that at mid-infrared wavelengths[12]. Finally, the enhanced terahertz field in the vicinity of metallic tips could serve as a quasi-static field capable of influencing intense laser-induced electron rescattering processes in a surrounding atomic or molecular gas, potentially increasing the high-energy cutoff or photon yield in high harmonic generation[38–40] or eliminating multiple electron returns that complicate the interpretation of recollision imaging measurements[9–11].

Given this motivation, we have investigated electron emission from tungsten wire tips that have been exposed to intense, single-cycle, terahertz pulses. We observe electron energies in excess of 5 keV, more than an order of magnitude greater than those previously reported in the near-infrared, infrared and terahertz regimes, implying the presence of acceleration gradients as large as 100 GeV m$^{-1}$. It is likely that we have produced electrons with energies in excess of 10 keV using the highest terahertz fields available in our laboratory, but these have yet to be confirmed due to voltage limitations in our current spectrometer. The maximum electron energies are directly proportional to the peak terahertz field, reflecting large local field enhancement factors that are inversely proportional to the tip radius, $R$, and greater than 3,000 for the sharpest tips. We have measured the emission from tips whose radii (and local field enhancement factors) differ by nearly two orders of magnitude and, interestingly, find that the maximum electron energy varies by only a factor of approximately two at any incident terahertz field strength. This result can be understood by modelling the sub-cycle energy transfer in the locally enhanced field of the wire tip.

## Results

**Characteristics of the electron field emission.** In the experiments, electrochemically etched tungsten wire tips (for example, see Fig. 1a) are exposed, in vacuum, to intense single-cycle terahertz pulses. As illustrated in Fig. 2a, electrons emitted from an electrically grounded tip, which is mounted on a static-field-free region of a retarding potential spectrometer, are collected by a micro-channel plate (MCP) detector. The electron energy spectrum is measured by varying the magnitude of a retarding potential placed on the entrance to the MCP. For direct comparison of the emission characteristics of different tips under identical conditions, several tips with different radii and/or cone angles can be simultaneously mounted on the spectrometer. The 6 mm separation between adjacent wires is large compared with the ~2 mm terahertz focus, ensuring emission from one, and only one, tip in any given measurement. A precision translation stage is used to move individual tips into and out of the terahertz focus, enabling measurements with different tips under identical experimental conditions. In most cases, the tungsten wires are mounted parallel to the spectrometer axis with their tips pointed directly toward the MCP, and the positive half-cycle of the terahertz field is directed from the wire tip to the MCP. In this standard configuration, electron emission cannot occur during the positive half-cycle of the single-cycle terahertz field (see Fig. 2b). Instead, the negative half-cycle is responsible for the field emission and subsequent energy transfer.

The terahertz pulses used in the experiments are characterized via photoelectron streaking in the same vacuum apparatus and spectrometer used for the field emission experiments[41]. The

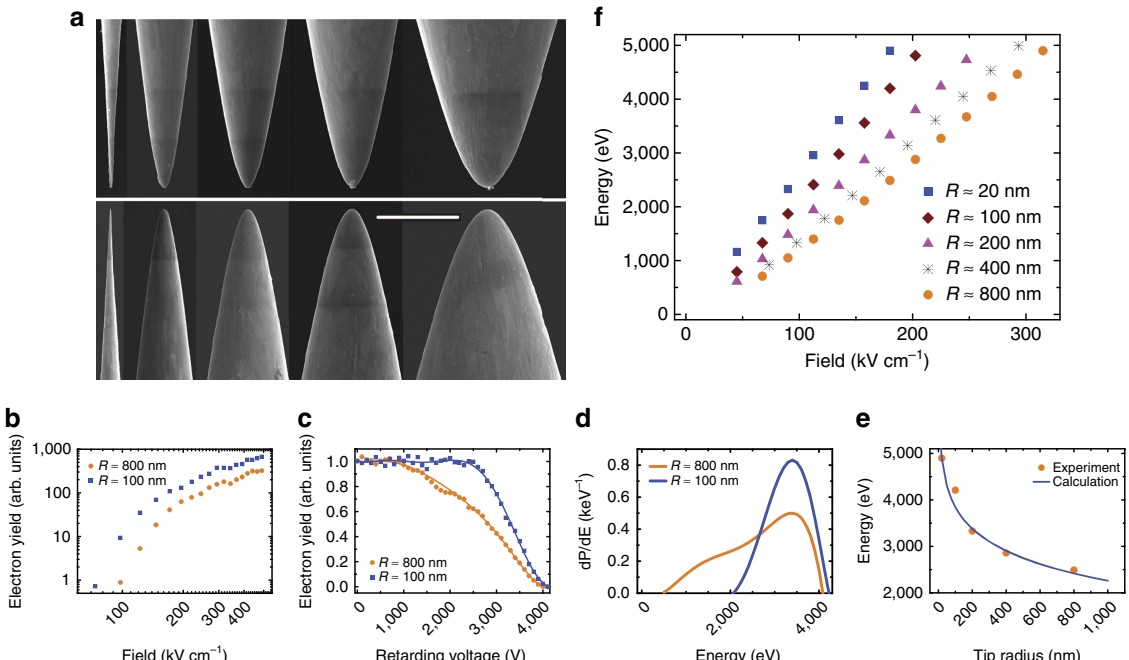

**Figure 1 | Terahertz-induced field emission from tips of different radii. (a)** SEM images of the five tungsten tips used to produce the data shown in **b–f**. The lower panel shows tips before terahertz exposure. The upper panel shows tips, which have been exposed to over 100,000 shots with a peak terahertz field of 450 kV cm$^{-1}$. From left to right, the tip radii are $R \simeq 20$, 100, 200, 400 and 800 nm. The white horizontal scale bar has a length of 5 µm. The angular orientations of the tips in the upper and lower panes are not identical. Aside from small carbonaceous patches that form due to the attraction of gaseous background molecules to the transiently charged tips, only the sharpest tip shows any apparent terahertz-induced modification of the tungsten surface. **(b)** Electron yield as a function of peak terahertz field strength for tips with radii of $R \simeq 100$ nm (blue) and $R \simeq 800$ nm (orange). **(c)** Electron yield versus retarding voltage (data points) and smooth (eighth-order polynomial) fits to those data (solid lines) for the $R \simeq 100$ nm (blue) and $R \simeq 800$ nm (orange) tips. The data were collected using different terahertz field strengths ($\sim 180$ and $\sim 270$ kV cm$^{-1}$ for the 100 and 800 nm tips, respectively) to generate spectra with similar cutoff energies, $\sim 4$ keV. **(d)** Energy distributions derived from the smooth fits to the yield data shown in **c**. **(e)** Measured (filled circles) and calculated (solid curve) electron cutoff energy as a function of tip radius for a fixed incident terahertz field strength $F \simeq 180$ kV cm$^{-1}$. **(f)** High-energy cutoff of the electron energy spectrum as a function of incident terahertz field strength, for tips with $R \simeq 20$ nm (filled blue squares), 100 nm (filled brown diamonds), 200 nm (filled magenta triangles), 400 nm (black asterisks) and 800 nm (filled orange circles).

pulses are passively carrier-envelope phase stable, with single-cycle sine-like waveforms, a central frequency $\nu \simeq 0.15$ THz, a maximum field strength (at focus) of $\sim 450$ kV cm$^{-1}$ and a field amplitude asymmetry of $\sim 2/3$ between the positive and negative half cycles. Figure 2b shows the terahertz waveform and its spectrum. For additional experimental details, see 'Methods' section.

We have observed electron emission from a variety of tungsten tips having different cone angles and tip radii, 10 nm $< R <$ 1,000 nm (see for example, Fig. 1). The majority of the tips are relatively blunt in comparison with the sharper (1–20 nm) tips that are typically used in the studies of ultra-fast, nano-tip field emission[12–15,18,20–22,30–32,35–37]. Even so, as shown in Figs 1 and 3, electron emission is observed for incident terahertz fields as low as 40 kV cm$^{-1}$. In all cases, we find that the most probable and maximum electron energies scale linearly with the peak terahertz field.

In an attempt to characterize any tip degradation due to the high-energy electron emission and the large enhanced fields at the tip surfaces, we first configured the spectrometer to search for ion emission, but none was observed. As a further check, scanning electron microscope (SEM) images of the tips were taken before and after their exposure to the terahertz field. Each of the tips pictured in Fig. 1a was exposed to a peak terahertz field of 450 kV cm$^{-1}$ for more than 100,000 shots prior to making the measurements shown. Close inspection of Fig. 1a reveals the presence of small carbonaceous patches (confirmed by elemental analysis in the SEM) on some of the tips, post exposure. These

apparently form when background gas molecules are attracted to the tip when it becomes transiently charged in the terahertz field. We have confirmed, however, by comparing the emission from similarly shaped tips with and without such patches, that the electron energies are not influenced by their presence. More interestingly, the post-exposure images indicate that, among the five tips shown, only the 20 nm tip suffered any notable modification of the actual tungsten surface. Although this tip remained intact with comparable tip radius, a very thin sheet of material appears to have been projected outward some tens of nm in a plane, extending on either side of the cone, over a distance of $\sim 1$ µm from the apex. Apparently, this tip was exposed to conditions near the threshold for significant surface modification. That said, the 450 kV cm$^{-1}$ field to which the 20 nm tip was exposed was more than twice that needed to produce 5 keV electrons (see Fig. 1f). Also, slightly blunter tips, with radii of $\sim 40$ nm showed no significant tungsten surface modification after exposure to more than half a million shots with a field that was larger than that required to produce 5 keV electrons. For all the tips studied, the maximum electron energies produced for a given incident field were independent of net exposure time, with some tips taking more than 2 million high-field shots.

The data in Figs 1 and 3 were obtained in a parallel geometry in which the terahertz polarization, the tungsten wire and the spectrometer axis are co-linear. No electrons are detected when the wire is oriented perpendicular to the spectrometer axis, indicating that the emission is confined to an acute solid angle centred along the wire axis. In addition, comparison of the

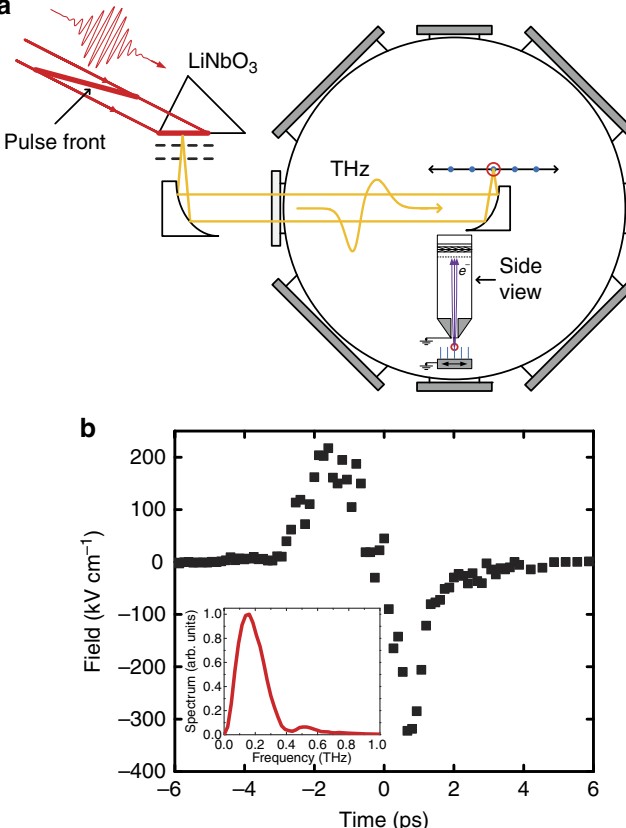

**Figure 2 | Experimental apparatus schematic and terahertz field characteristics.** (**a**) Terahertz pulses are produced via optical rectification, at a 15 Hz repetition rate, using 790 nm, 150 fs and 20 mJ pump laser pulses in a tilted pulse-front configuration. The terahertz pulses pass through two wire-grid polarizers, which serve as a variable attenuator. Two off-axis parabolic mirrors focus the terahertz radiation on a single wire tip (identified by a red circle in the main figure and side view inset) positioned at the centre of a retarding potential electron energy analyser (see side view inset) near the centre of the vacuum chamber. A precision translation stage allows the positioning of different tips in the terahertz focus, allowing us to maintain identical experimental conditions for measurements from different tips without breaking vacuum. (**b**) Terahertz electric field (main figure) obtained from photoelectron streaking measurements[41] and corresponding frequency spectrum (inset) obtained by Fourier transforming the time-domain data shown in the main figure.

electron yield as a function of terahertz polarization direction (at fixed terahertz field strength), with the yield obtained as a function of terahertz field strength in the parallel wire-field geometry, confirms that the emission depends only on the component of the applied field parallel to the wire axis. Therefore, in the remainder of the paper, we focus on results obtained in the parallel wire-field configuration.

In general, at low terahertz fields, the electron current versus field exhibits the expected exponential form for Fowler–Nordheim tunnelling[42]. However, as shown in Fig. 1b, at higher fields, the yields saturate and exhibit a more complex field-dependent behaviour, perhaps due in part to non-adiabatic energy transfer to the electrons within the wire. That said, our principal interest in this communication is not on the mechanism responsible for liberating electrons from the tip, but rather on the process through which those electrons obtain such large kinetic energies from the field.

**Electron energy versus incident field strength.** We first consider how the energies of the emitted electrons from a single tip depend on the applied terahertz field. As a typical example, Fig. 3a shows the energy distributions of the electrons emitted from an $R \simeq 130$ nm tip at five different terahertz fields. The maximum energies observed are plotted as a function of terahertz field strength in Fig. 3b. The general shape of each distribution, a well-defined peak with a sharp high-energy cutoff, is similar to those previously reported for terahertz-induced field emission[36], and the linear scaling of the cutoff energy versus incident field is a signature of energy transfer in the strong-field, sub-cycle regime[12]. The features in Fig. 3a have low energy tails that are less pronounced than those reported by Herink et al.[36], but the scaled widths (full-width at half-maximum divided by peak energy), which are roughly independent of terahertz field, are comparable to theirs. More interestingly, however, the distributions shown in Fig. 3a have peak energies that are $\sim 30$ times larger than those measured previously at nearly identical incident terahertz field strength[36]. Apparently, the substantially larger electron energies can be attributed to the lower terahertz frequency used in the current experiment ($\simeq 0.15$ THz versus $\simeq 1.1$ THz), and/or the larger tip radius ($\simeq 130$ nm versus $\simeq 10$ nm). The maximum energies we observe are also more than an order of magnitude larger than those produced with mid-infrared pumping at intensities that are five orders of magnitude higher[12].

**Dependence of maximum electron energy on tip radius.** To better understand the mechanism responsible for the

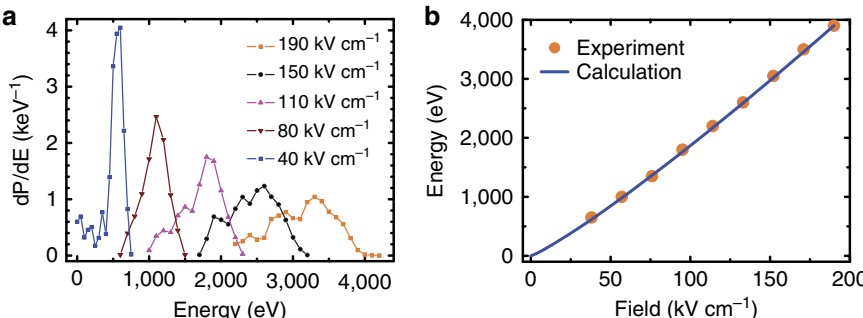

**Figure 3 | Energy distributions versus incident terahertz field strength.** (**a**) Energy distribution of electrons emitted from a tip with $R \simeq 130$ nm for different incident terahertz fields: 40 kV cm$^{-1}$ (blue); 80 kV cm$^{-1}$ (brown); 110 kV cm$^{-1}$ (magenta); 150 kV cm$^{-1}$ (black); and 190 kV cm$^{-1}$ (orange). Lines connecting data points are provided to guide the eye. The total yield at each field has been normalized to unity. (**b**) High-energy cutoff of electron energy spectra as a function of incident terahertz field strength for the same tip used to produce the data shown in **a**. Solid orange circles are data and the solid blue line is the result of the simulation described in the text.

high-electron energies produced in our experiments, we have also examined the electron emission as a function of tip radius and cone angle. Figure 1a shows SEM images of five tips with radii $R \simeq 20$, 100, 200, 400 and 800 nm. Figure 1e shows the high-energy cutoff of the electron distributions from each tip as a function of incident terahertz field. For sufficiently high-incident fields, electrons with energies exceeding 5 keV are produced from all of the tips. However, due to limitations of the retarding potential spectrometer, only cutoff energies $< 5$ keV can be measured. Figure 1d shows the variation in cutoff energy with tip radius for a fixed incident terahertz field of 180 kV cm$^{-1}$. Naively, one might expect that the highest energy electrons would be produced from the sharpest tips, due to the larger local field enhancement. This is indeed the case, but perhaps not to the degree anticipated. Notably, we observe no significant difference in the cutoff energies from tips with approximately the same tip radius but with cone angles differing by as much as 60%. Moreover, for a given incident field, we observe only a factor of two decrease in the maximum electron energy when the tip radius is increased by a factor of 40 (Fig. 1b,c).

## Discussion

The linear, rather than quadratic, scaling of the maximum electron energy with incident field strength indicates that the underlying physical mechanism for the energy transfer is substantially different from that predicted by the simple man's model for electron emission in the presence of an oscillating field. Indeed, our observations are consistent with a different model[12] in which the energy transfer to the electron is confined to a fraction of the field cycle. As such, it depends on the magnitude of the locally enhanced field at the instant of emission, rather than on the vector potential (which reflects the time integral of the field after emission).

The sub-cycle energy transfer process as it applies to our measurements can be understood as follows. When a tip with radius of curvature, $R$, is exposed to a terahertz field, the induced charge at the surface of the tip modifies the local field surrounding it. A discrete series-$RLC$ circuit model indicates that for tungsten tips with $R > 10$ nm driven at wavelengths comparable to, or greater than, that used in our experiments, the induced surface charge (and field) at the tip oscillate approximately in-phase with the incident field (see 'Methods' section). As a result, the field enhancement factor $\gamma$ can be computed in the static field limit. In contrast, at higher frequencies, the decrease in the capacitive reactance and increase in the inductive reactance can result in a non-negligible phase-shift between the incident and enhanced fields, potentially limiting the local enhancement for short drive pulses. For our measurements, the surface field at any instant can be written as, $F = \gamma F_0$, where $F_0$ is the incident terahertz field. An electron, ejected from the surface, rapidly accelerates away from the tip in the locally enhanced field. The local field drops rapidly to $F_0$ as the electron travels a distance comparable, and proportional, to $R$ (ref. 43). The electron leaves the vicinity of the tip in a time much less than the terahertz period, so the incident field remains roughly constant during that time. The electron's energy gain is equal to the work done by the enhanced local field which, in turn, is approximately equal to the average electric force $\frac{1}{2}e\gamma F_0$ on an electron with charge, $e$, multiplied by the distance, $\sim R$, over which the enhanced field decays.

$$\Delta E = \frac{1}{2} e \gamma F_0 R. \qquad (1)$$

For $\gamma \gg 1$, this local energy transfer is much greater than that acquired by the electron (up to eight times the pondermotive energy) as it moves in the bare, single-cycle terahertz field far

from the tip[41]. According to equation 1, the final energy, $\Delta E$, of an ejected electron will be proportional to the magnitude of the incident field at the instant of its emission, with the maximum electron energy proportional to the peak terahertz field. This prediction is in excellent agreement with our measurements (see Figs 1b and 4b).

The sub-cycle energy transfer model also explains the absence of a strong dependence of $\Delta E$ on $R$. To see this, $\gamma$ must be expressed in terms of relevant geometric parameters. For a hemisphere-capped cylinder of length $L$ and radius $R$ in a static field, $\gamma$ is approximately proportional to $\sim L/R$ (ref. 44). However, in the presence of terahertz radiation of wavelength $\lambda$, retardation effects will limit the effective length of the emitter to $L < \lambda/2$ (ref. 45). As a result, equation 1 reduces to

$$\Delta E \sim \frac{1}{4} e F_0 \lambda. \qquad (2)$$

Beyond the linear dependence on the applied field, equation 2 predicts an energy transfer that is directly proportional to the terahertz wavelength, and independent of tip radius. Electrons ejected from sharper tips experience a larger enhanced field as compared with blunter tips, but the field decays over a proportionally shorter distance. For a peak terahertz field strength $F_0 = 180$ kV cm$^{-1}$ and a terahertz wavelength $\lambda = 1$ mm, equation 2 obtains a maximum energy transfer $\Delta E \sim 4.5$ keV, in reasonable agreement with the measurements, $2,500$ eV $\leq \Delta E \leq 5,000$ eV, for 800 nm $\geq R \geq 20$ nm. Although the measured energy transfer does show some $R$-dependence, that dependence is small, with a variation in $\Delta E$ of only a factor of 2 for tip radii differing by a factor of 40.

To reproduce the (small) observed $R$-dependence in $\Delta E$, we augment the zeroth-order model presented above. Specifically, approximate analytic forms for both the time-dependent tunnelling probability and the spatial distribution of the enhanced field are used to compute the energy distributions of the detected electrons for one-dimensional motion along the tip axis, $z$. Rather than attempting to precisely model the geometry of the tips and compute the enhanced field via finite-difference techniques, we employ a generalized form for the field enhancement factor at the tip apex[44], $\gamma = \beta(\lambda/R)^{\alpha}$, where $\beta$ and $\alpha$ are constants to be determined. The Fowler–Nordheim tunnelling formula[42] is used to compute the rate at which electrons are emitted (with zero initial energy) into the enhanced field. Non-relativistic, classical equations of motion are then used to determine the final energy of electrons emitted at each time, $t$, using the known waveform of the incident field and the expression, $F(z, t) = F(0, t)/(1 + 2z/R)$ (ref. 46), for the decay of the electric near-field along the axis of a hyperboloidal tip. The constants $\alpha$ and $\beta$ that appear in the field enhancement factor are varied to obtain the best agreement with experiment.

As shown in Figs 1c and 4b, the simulation accurately reproduces the maximum electron energies observed as a function of peak terahertz field and tip radius for $\beta = 0.06$ and $\alpha = 1$. The calculations indicate that the (slight) decrease in $\Delta E$ with increasing radius is due to the greater time that the electron spends in the enhanced field, reducing the accuracy of the instantaneous field approximation used to obtain equations 1 and 2. The assignment $\alpha = 1$ agrees with our initial assumption that $\gamma$ is inversely proportional to $R$, while $\beta = 0.06$ indicates that the effective tip length is considerably smaller than $\lambda/2$, consistent with previous calculations for incident 800 nm fields[47]. The calculated maximum energies are not changed if the Fowler–Nordheim tunnelling rate is modified to simulate the yield saturation that is observed at the highest terahertz fields in the experiments (see Fig. 1b).

One feature of the data that is not satisfactorily reproduced by the simulations is the width of the electron energy distributions (for example, Fig. 3a). Calculations performed using a saturated ionization rate to simulate the field-dependent yields observed in experiments do predict non-negligible electron emission over a wider range of times (phases) within the enhanced field as compared with calculations assuming pure Fowler–Nordheim tunnelling. However, the principal effect on the energy distributions is an increase in the amplitude of the low energy tail, with little change to the width of the primary peak. Thus, it is unlikely that the variation in emission times is primarily responsible for the measured bandwidth. Instead, much of the observed widths may be due to non-negligible interactions between electrons in-flight. However, such space-charge effects do not explain the broadening of the electron energy distributions with increasing $R$. Figure 1c shows the energy distributions of the electrons emitted from tips with $R \simeq 100$ and $R \simeq 800$ nm at peak terahertz fields of 180 and 270 kV cm$^{-1}$, respectively. The distributions have approximately the same high-energy cutoffs, but substantially different widths. Notably, the electrons are emitted over a much larger area with the blunt tip, and the total yield is lower than that from the sharper tip (see Fig. 1b). Consequently, the density of electrons leaving the blunt tip is more two orders of magnitude smaller than that from the sharp one, arguing against space-charge interactions as the source of additional broadening. Simulations indicate that the breakdown in the instantaneous field approximation with increasing tip size can produce some energy broadening, but not to the extent observed. Another possibility is an increase in the number of detected electrons that are emitted at lower energies from the sides (rather than the apex) of the larger radius tips, which also have larger cone angles. A simulation of the many-electron propagation problem for the precise emitter and spectrometer geometries may be needed to fully understand the details of the energy distributions. Unfortunately, such a calculation is beyond our current capabilities.

In summary, we have used strong single-cycle terahertz fields to induce electron emission from tungsten nano-tips. Electrons with energies easily exceeding 5 keV have been detected, substantially greater than previously observed, using terahertz, infrared or near-infrared driving. The maximum electron energies are proportional to the peak terahertz field and roughly independent of tip radius, in good agreement with the predictions of a sub-cycle energy transfer model. Accordingly, the longer wavelength used in this work, as compared with previous photoemission studies, appears to be the primary factor responsible for the significantly higher electron energies. Following a full characterization of their temporal and angular distributions, these isolated high-energy electron bursts might serve as sources[48,49] to enable a variety of applications, including time-resolved electron diffraction[37]. In addition, in this long wavelength regime, the enhanced terahertz fields in the vicinity of tips with radii, $R > 1$ μm, while insufficient to induce electron emission, could be employed to explore the influence of large quasi-static fields on high harmonic generation, attosecond pulse generation and other phenomena based on electron rescattering in an intense laser focused near the tip[38–40].

## Methods

**Tungsten tip preparation.** The tungsten wire tips are prepared via electrochemical etching: tungsten wire (99.95% purity, 0.25 mm diameter), graphite rod (99.995% purity, 6 mm diameter) and 3 mol l$^{-1}$ NaOH solution (NaOH dissolved in distilled water) serve as the anode, cathode and electrolyte, respectively. The anode immersion depth is 2 mm and etching proceeds at an applied voltage of 4 V. The radius of the etched tip is controlled by varying the post-etching time, that is, the time delay between the drop-off of the submerged portion of the wire and the switch-off of the etching voltage.

**Terahertz pulse generation and characterization.** The terahertz radiation is generated in LiNbO$_3$ via tilted-pulse-front-pumping optical rectification of 150 fs, 790 nm and 20 mJ pulses from a Ti:sapphire laser[50,51]. The temporal profile of the radiation is measured *in situ* by photoelectron streaking[3,41,52]. The spatial profile of the terahertz beam, at its focus, is measured using a pyroelectric detector. The field strength is calibrated using terahertz-induced, strong-field ionization of Na Rydberg atoms[41]. Further details regarding the terahertz characterization can be found in ref. 41.

**Experimental set-up.** A schematic of the experimental apparatus is shown in Fig. 2a. The background pressure of the vacuum chamber is maintained at $\sim 3 \times 10^{-7}$ Torr, and reduced, during data collection, to $\sim 1 \times 10^{-7}$ Torr using a liquid nitrogen cold trap. After etching, the tungsten tips are imaged using a SEM, and those with desired sizes are selected and immediately mounted in the experimental chamber to inhibit oxidation. The tips are mounted in a grounded conducting holder that is rigidly attached to a mechanical translation stage. The holder can accommodate up to five wires, mounted in a line with $\sim 6$ mm gaps between adjacent tips. The gap between wires is much larger than the $\sim 2$ mm focal diameter of the terahertz beam, ensuring that only a single tip is exposed to the field at any given time. The terahertz beam is focused on the wire target using two off-axis parabolic mirrors. The intensity and polarization of the terahertz beam are controlled using a variable attenuator composed of two wire-grid polarizers. The maximum possible retarding potential in the energy analyser is 5 kV.

**Series-*RLC* circuit model.** We treat the nano-tips as discrete, driven series-*RLC* circuits to determine the phase of the charge-enhanced field at the tip, relative to that of the terahertz drive[31]. The capacitance, $\mathcal{C} = 0.5, 0.6, 0.7, 0.8$ and $0.9$ fF, of the tips with radii, $R = 20, 100, 200, 400$ and $800$ nm, respectively, is estimated by treating each tip as a hemisphere-capped cylindrical rod of radius $R$ and an effective length, $L = 0.06\lambda$, equal to that used in calculating the enhancement factor $\gamma$ (ref. 53). The inductance, $\mathcal{L} = 100, 75, 70, 60$ and $50$ pH, for $R = 20, 100, 200, 400$ and $800$ nm, respectively, is estimated by treating each tip as a cylindrical rod of radius $R$ and effective length $L = 0.06\lambda$ (ref. 54). Despite the actual variation in $R$ along the conical tips, these simplified models should provide reasonable results given the very weak dependence of $\mathcal{C}$ and $\mathcal{L}$ on $R$. Indeed, the values of $\mathcal{C}$ and $\mathcal{L}$ reported previously for $R = 40$ nm gold tips[31] are in good agreement with those computed here. The resonant frequencies, $f_0 = 1/(2\pi\sqrt{\mathcal{LC}})$, range from $\sim 0.7$ to $0.8$ terahertz for the five tips, well above the high-frequency cutoff of the applied terahertz field (see Fig. 2b). This supports our use of the static field limit in computing the enhanced field at the tip. Further justification for this DC approximation is obtained by determining the standard *RLC* phase-shift $\phi$ between the maximum charge on the (capacitive) tip and the maximum of the incident terahertz field. To do so, we use the capacitance and inductance values (above) in combination with estimates of the effective resistance $\mathcal{R}$ of each tip. We assume the resistance is equal to that of a truncated conical conducting shell of effective length $L = 0.06\lambda$ and a thickness equal to the tungsten skin depth, $\delta$. We assume a frequency $f \sim 300$ GHz, based on the duration of the negative half-cycle of the field (see Fig. 2b), giving $\delta \sim 200$ nm. The resistance, $\mathcal{R} \simeq \rho \ln(r_b/R)/(\pi\delta\tan\theta)$, where $\rho$ is the resistivity, is more sensitive to the cone half-angle $\theta$ than to the tip radius $R$ or the radius of the cone base, $r_b$. The resistances fall in the range $0.8\,\Omega \leq \mathcal{R} \leq 7\,\Omega$ for the five tips used in the measurements, yielding phase-angle values for all tips, $\phi < 1°$.

**Data availability.** The data that support the findings of this study are available from the corresponding authors upon request.

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

## Acknowledgements

It is a pleasure to acknowledge helpful conversations with M. Krüger and M.F. Kling. This work has been supported by the U.S. Department of Energy, Office of Science, Office of Basic Energy Sciences, Award # DE-FG02-00ER15053 (S.L.) and Award # DE-SC0012462 (R.R.J.).

## Author contributions

S.L. set up and performed the experiments, analysed the experimental data and developed and performed the simulations. R.R.J. conceived the initial experiments, and provided oversight and guidance in the performance of the apparatus modifications, data collection, data analysis and simulations. Both authors contributed equally to the manuscript.

## Additional information

**Competing financial interests:** The authors declare no competing financial interests.

**Publisher's note**: 

