## [Peer review file · Nature Communications]

Reviewers' comments:

Reviewer #1 (Remarks to the Author):

The paper is timely given the rapid progress in strong field THz generation and the progress in strong field physics at nano-tips. The paper is also well written. The high electron energies generated directly from the tips using long wavelength (THz) radiation is novel, however also has serious limitations in the quality of electron bunches it provides as is discussed in more detail below.

The electron source demonstrated is based on physics where the THz field is enhanced by a factor of 1000 or more at the tips according to the authors. This gives fields in excess of 200 GeV/m at the tip surface and there arises the question of tip damage which is not addressed at all in the paper. How long can these tips be operated under those conditions? 500 GV/m is the field that holds the electron on the 1s hydrogen orbit. One should show SEM figures of the tips after a given number of shots.

It may be long enough to measure the electron statistic for the tip emission, but it may not be enough for any kind of electron microscopy as mentioned at the end of the paper as a possible application of this kind of electron gun.

Also nothing is said about the number of electrons generated per shot as a function of tip radius R and field strength. This needs to be added. It would be great also if the emittance of the electron bunches can be addressed.

Beyond possible damage, which is not discussed here at all, the demonstrated source is severely limited in its applicability. At such high on-tip fields, emission must occur over most of the accelerating quarter of a THz cycle, which is a duration of about 1 ps. Thus the generated electron pulses are not particularly short. The broad energy electron spectra observed of about 20% of its central energy or more is indicative and a consequence of that. The emittance of small tips is typically equal to the electron tip size. The smallest here is 20 nm, which is not particularly small and therefore the emittance not particularly high, which limits its application for time resolved electron diffraction and microscopy considerably. The results achieved with larger tips are even less interesting. Small DC-guns as developed by many groups now, for example those of the Miller group, U. of Toronto, outperform this gun considerably, providing 100 fC bunches with <100 fs duration and emittance probably also approaching 20 nm rad. Those guns typically operate at 1 kHz repletion rate or higher and not 15 Hz as demonstrated here.

For these reasons also very interesting, I cannot recommend publication in Nature Communications.

Reviewer #2 (Remarks to the Author):

This interesting manuscript describes an experimental study of THz-induced field emission and electron acceleration from metallic nanotips. Kinetic energies up to 5 keV are observed, and possibly higher energies are claimed (but could not be determined experimentally).

A systematic study of reachable kinetic energies as a function of radius of curvature of the tip is conducted, and indicates a weak dependence of final kinetic energy. The observations are explained within a quasi-instantaneous acceleration model.

The work is very appealing and on a subject of high current interest. The experiments and analysis are carefully designed and conducted. The paper is generally well-written and comprehensible, and it gives proper reference to prior literature. The demonstration of keV kinetic energies from THz field emission is a significant achievement, which will find general attention.

I expect that I will suggest publication of this very nice work after the authors consider the following comments in a revision:

1) The figure composition and preparation could use some additional attention. Some of the graphs are partially redundant (1b and 4b) or contain similar information. Figures 2 and 3 contain relatively little information and should probably be combined.

2) I believe a plot of the measured field-dependent currents should be provided, together with a discussion. How do these currents compare for different tip radii and similar kinetic energies?

3) I did not find a very convincing discussion of the origin of the very broad kinetic energy spectra observed. Is this due to a broadened range of emission phases? Is this consistent with the observed nonlinearities?

4) The authors claim that the enhanced field is in phase with the incident field. What is the reasoning behind this assumption? Which capacitance and inductance would they assume for these structures, and is this consistent with the literature?

5) The peak fields mentioned (250 GV/m) are above the (static) threshold for field ion evaporation. Have the authors checked the tip shapes after the experiments, and have they observed degradation effects? This should probably be discussed.

6) The authors state: "Examples of related phenomena include ponderomotive acceleration and electron rescattering of ejected electrons [15, 25], carrier-envelope phase (CEP) effects [15, 25], and extreme-ultraviolet (XUV) high harmonic generation (HHG) [27-29]."

Ref. [29] shows that HHG is not the cause of the XUV emission in these works, but that the emission is rather caused by incoherent fluorescence (see also Sivilis et al., Phys. Rev. Lett. (2013) regarding Ref. [28], and a Comment on Ref. [27]). I would suggest to replace "extreme-ultraviolet (XUV) high harmonic generation (HHG)" with something like "strong-field-ionization induced XUV fluorescence".

7) The authors may want to also refer to early works on field emission in microwave fields (Charbonnier et al., Proc. IEEE (1963) or Fursey, J. Vac. Sci. Techn. B (1995)).

Reviewer #3 (Remarks to the Author):

The paper by Li and Jones is a beautiful and careful study of field-enhanced electron emission from tungsten nano-tips in a THz field. This subject has been an area of considerable interest both from a fundamental and applied perspective. Recent reports have studied the emission using pulses that varied from near-visible to long-wavelength (10 micron). In the current paper, wavelengths (200 microns) that are 10-times longer than previous reports are employed. The results are spectacular, electrons are emitted in excess of 5 keV energy even though the photon energy is less than a millivolt. These observed energies are order of magnitude(s) higher than all previous reports. The reason appears to be linked to the longer wavelength which results in a near-static field as the electron traverses the enhancement region. Enhancement values of 3000 are reported. The study systematically investigates the behavior as a function of field strength and nano-tip size. A simple classical model in an exponentially decaying field nicely reproduces the results.

This is a very nice piece of work which will have broad interest in the photonics and strong-field community. I strongly recommend publication in Nature Comm.

One minor typo on pg 2, line 41: ponderomotive is misspelled

Response to Reviewer #1:

Reviewer #1: "The paper is timely given the rapid progress in strong field THz generation and the progress in strong field physics at nano-tips. The paper is also well written."

Authors: We appreciate the positive comments.

Reviewer #1: "The high electron energies generated directly from the tips using long wavelength (THz) radiation is novel, however also has serious limitations in the quality of electron bunches it provides as is discussed in more detail below. The electron source demonstrated is based on physics where the THz field is enhanced by a factor of 1000 or more at the tips according to the authors. This gives fields in excess of 200 GeV/m at the tip surface and there arises the question of tip damage which is not addressed at all in the paper. How long can these tips be operated under those conditions? 500 GV/m is the field that holds the electron on the 1s hydrogen orbit. One should show SEM figures of the tips after a given number of shots. It may be long enough to measure the electron statistic for the tip emission, but it may not be enough for any kind of electron microscopy as mentioned at the end of the paper as a possible application of this kind of electron gun."

Authors: As detailed in the manuscript, and in our responses below, we contend that high-energy electron bursts produced by THz irradiation of nano-tips could be useful for various applications. First, regarding the field at the tip surface, the original manuscript stated that the observed electron energies implied an acceleration gradient as large as 250 GeV/m. This number was based on a rough estimate of the distance travelled (on the order of the tip radius, R) by the electrons from the finest tip ($R \sim 20$ nm) to achieve the maximum energy of 5keV. We make a more careful estimate in the revised manuscript, 100 GeV/m, based on the inferred maximum enhanced field at the finest tip. Indeed, this is still a very large field and in regard to the questions of tip damage, we did make subsidiary measurements to search for, and characterize, it. Specifically:

- i) We used the time-of-flight spectrometer to search for positive or negative ions ejected from the tip, but found no evidence for any. We note, however, that ions formed by electron collisions with background gas could be observed when air was leaked into the chamber at pressures in the mid 10^{-5} Torr range and above.
- ii) We did not observe any notable reduction in the maximum energies produced by any tips with increased exposure time. The longest net exposure time was approximately 50 hours at, or near, the maximum possible THz field (> 2 million high-field shots). The set of 5 tips with different radii shown in Fig. 1 were each exposed to the full THz field ($F_{\text{max}} \sim 450$ kV/cm) for two hours ($> 100,000$ shots) prior to collecting the data shown.

iii) Most of the tips we have studied were imaged with an SEM both before and after THz exposure, and we now include both before- and after-exposure images in Fig. 1. For the finest tip, $R = 20$ nm, we are able to cross the threshold for tip modification at the very highest enhanced fields. In Fig. 1, only the 20 nm tip shows any modification of the tungsten surface. In that case, the main tip has survived with comparable radius, but a very thin sheet of material has apparently been projected outward some 10s of nm in a plane, extending on either side of the cone, over a distance of ~ 1 micron. It is worth noting, however, that this tip was exposed ($> 100,000$ shots) to a field that was 2.3x greater than required to produce 5 keV electrons. Conversely, a 40 nm tip not shown, which was exposed to $> 1/2$ million shots with a field that was larger than needed to produce 5 keV electrons, showed no significant surface modification. In addition to tungsten surface modification, small patches of carbonaceous material (confirmed by elemental analysis in the SEM) can be seen on some of the tips after removal from the vacuum system. By comparing the emission from tips of comparable size and shape, with and without these patches, we have confirmed that the electron energies are not influenced by this growth which apparently forms when background hydrocarbons are attracted to the tip when it becomes charged in the presence of the THz field.

Changes to the Manuscript:

a) The maximum estimated acceleration gradient is given as 100 GeV/m in line 76.

b) The discussion of tip exposure times, our search for ejected ions, and our observation of limited tip modification now appears in lines 118-140.

c) Pre- and post-exposure tip images now appear in Fig. 1a. The Fig. 1 legend has been modified accordingly.

Reviewer #1: "Also nothing is said about the number of electrons generated per shot as a function of tip radius R and field strength. This needs to be added. It would be great also if the emittance of the electron bunches can be addressed."

Authors: The original manuscript contained a brief discussion of the dependence of the observed electron current on the applied field. The revised manuscript includes a new plot (Fig. 1b) which shows the detected electron yield as a function of applied THz field strength for a typical (100 nm) and blunt (800 nm) tip. The yield axis actually gives the estimated number of electrons per shot. However, the axis scale is given in "arb. units" as the detector calibration changes over time and is only approximate. We cannot measure the angular distribution of the electron bursts without a substantial modification to the apparatus and, therefore, do not have a value for the emittance.

Changes to the Manuscript:

a) Fig. 1b has been added to the manuscript resulting in a relabeling of the other panels

b) The Fig. 1 legend has been modified accordingly. We refer to this figure when discussing the dependence of the observed electron current on the applied field in lines 151, 260, and 276.

Reviewer #1: "Beyond possible damage, which is not discussed here at all, the demonstrated source is severely limited in its applicability. At such high on-tip fields, emission must occur over most of the

accelerating quarter of a THz cycle, which is a duration of about 1 ps. Thus the generated electron pulses are not particularly short. The broad energy electron spectra observed of about 20% of its central energy or more is indicative and a consequence of that. The emittance of small tips is typically equal to the electron tip size. The smallest here is 20 nm, which is not particularly small and therefore the emittance not particularly high, which limits its application for time resolved electron diffraction and microscopy considerably. The results achieved with larger tips are even less interesting. Small DC-guns as developed by many groups now, for example those of the Miller group, U. of Toronto, outperform this gun considerably, providing 100 fC bunches with <100 fs duration and emittance probably also approaching 20 nm rad. Those guns typically operate at 1 kHz repetition rate or higher and not 15 Hz as demonstrated here."

Authors: As noted above, we now show (Fig. 1a) and discuss (lines 118-140) the fact that there is negligible tip modification except when the smallest tips are exposed to fields so large that the projected electron energies are well in excess of 5 keV. Our numerical simulations do indeed indicate that electrons can be emitted over intervals of several hundred femtoseconds. However, this does not imply that such a source is not useful. For example, Ropers' group has recently probed dynamics in polymer/graphene bilayers employing electron diffraction imaging with a nano-tipped based source which produces ~100 electron pulses with durations in the few picosecond range [Science 345, 200 (2014)]. The radius of the tip for that source was 50 nm, so tips with a smaller radius are not necessarily a requirement for useful sources. In addition, the spread in electron emission times, as well as temporal broadening due to electron-electron repulsion, does not preclude the possibility of obtaining very short pulses on target by employing a variety of pulse compression schemes. For example, to our knowledge, the <100 fs electron sources pioneered and used by Miller rely on rf compression techniques. For our source, an analogous, but better alternative to rf (if very short pulses are desired from a very compact source) would be THz-based pulse compression as recently demonstrated [Science 352, 429 (2016)]. Finally, the demonstrated 15 Hz duty cycle is not a fundamental limitation of the demonstrated technique, rather a current limitation in our laboratory. For the saturated LiNbO3 THz sources, the peak THz field is proportional to the square root of the incident pulse energy. We have shown that we can produce 5keV electrons with less than half of the peak THz field produced with our 20 mJ 15 Hz laser. Thus, it should be possible to produce 5 keV electrons using THz pulses created with a fairly standard 4 mJ, 100 fs, 1 kHz, Ti:Sapphire laser system.

Response to Reviewer #2:

Reviewer #2: "This interesting manuscript describes an experimental study of THz-induced field emission and electron acceleration from metallic nanotips. Kinetic energies up to 5 keV are observed, and possibly higher energies are claimed (but could not be determined experimentally). A systematic study of reachable kinetic energies as a function of radius of curvature of the tip is conducted, and indicates a weak dependence of final kinetic energy. The observations are explained within a quasi-instantaneous acceleration model. The work is very appealing and on a subject of high current interest. The experiments and analysis are carefully designed and conducted. The paper is generally well-written and comprehensible, and it gives proper reference to prior literature. The demonstration of keV kinetic energies from THz field emission is a significant achievement, which will find general attention. I expect that I will suggest publication of this very nice work after the authors consider the following comments in a revision:"

Authors: We appreciate the positive comments.

Reviewer #2: 1) The figure composition and preparation could use some additional attention. Some of the graphs are partially redundant (1b and 4b) or contain similar information. Figures 2 and 3 contain relatively little information and should probably be combined.

Authors: We have combined Figs. 2 and 3 as suggested. The figures initially labelled 1b and 4b (now 1e and 3b in the revised manuscript) do contain similar information, but for different tips and in different contexts. Figure 1e provides an explicit comparison of the maximum electron energy vs. incident THz field for the 5 tips for which additional emission data and images are shown in the other Fig. 1 panels. Figure 3b shows the same type of data, but for a different (single) tip, displaying the maximum energies at the THz field strengths for which full energy distributions are provided in Fig. 3a. Fig. 3b contains much less data than Fig. 1e, and as such is better suited to show the agreement between model calculations and experiment. In our opinion, adding five calculated curves to Fig. 1e, would make it overly cluttered, particularly at low fields. Therefore, we believe it is preferable to retain the separate plots.

Changes to the Manuscript:

Figs. 2 and 3 have been combined, resulting in a relabeling of Fig. 4 as Fig. 3. The Fig. 2 legend has been modified accordingly.

Reviewer #2: 2) I believe a plot of the measured field-dependent currents should be provided, together with a discussion. How do these currents compare for different tip radii and similar kinetic energies?

Authors: As described in the response to Reviewer #1, we have included data on the field-dependent electron yield for tips of different radii in Fig. 1b enabling the comparison the Reviewer #2 requests.

Reviewer #2: 3) I did not find a very convincing discussion of the origin of the very broad kinetic energy spectra observed. Is this due to a broadened range of emission phases? Is this consistent with the observed nonlinearities?

Authors: In the original manuscript, the width of the observed energy distributions was discussed in two different locations. First in the results section where we state that the fractional bandwidth is very comparable to that observed previously by other groups, but at significantly lower energy in those previous measurements. Second, in the second to last paragraph in the Discussion section where, among other things, we discuss the fact that the range of emission phases (or times) does not satisfactorily account for the width of the distributions. In the revised manuscript, we supply additional information in that Discussion paragraph. First, the simulations show, not unexpectedly, that the saturation of the electron yield with increasing field strength results in non-negligible emission over a wider range of times, leading to an increased low energy tail in the energy spectra. Second, the simulations show that, with increasing tip radius, the accuracy of the instantaneous field approximation is reduced, and the highest energy electrons are produced prior to the peak in the THz field cycle. This also leads to an increased yield in the low energy tail of the distributions, particularly for tips with larger radius. However, neither of these effects fully accounts for the observed width of the distributions which may be due to non-negligible interactions between electrons in-flight, and/or the collection of

electrons emitted at lower energies from the side (rather than the apex) of the tip. Accurately simulating both of these effects is beyond our current capability.

Changes to the Manuscript: We have added to the discussion of the potential sources of broadening in the observed energy distributions (lines 261-285).

Reviewer #2: 4) The authors claim that the enhanced field is in phase with the incident field. What is the reasoning behind this assumption? Which capacitance and inductance would they assume for these structures, and is this consistent with the literature?

Authors: As stated in the original manuscript, the assumption that the enhanced and incident fields are in phase is based on a series RLC circuit model. We have now included the details of the model in Methods under the sub-heading, "Series-RLC circuit model." The values we find for the inductance and capacitance are in good agreement with those reported for 40 nm gold tips in Nature Phys. 10, 432-436 (2014).

Changes to the Manuscript:

- i) We have added a new subsection "Series-RLC circuit model" to the Methods section.
- ii) References [53] and [54] have been added to support that subsection.

Reviewer #2: 5) The peak fields mentioned (250 GV/m) are above the (static) threshold for field ion evaporation. Have the authors checked the tip shapes after the experiments, and have they observed degradation effects? This should probably be discussed.

Authors: As described in the response to Reviewer #1, we have included the following in the revised manuscript:

- i) a more accurate value for the maximum fields generated at the tip surface (100 GV/m)
- ii) a set of post-exposure images of the tips showing negligible degradation for all but the sharpest (R=20nm) tip
- iii) a discussion of tip exposure times, our search for ejected ions, and our observation of limited tip modification

Reviewer #2: 6) The authors state: "Examples of related phenomena include pondermotive acceleration and electron rescattering of ejected electrons [15, 25], carrier-envelope phase (CEP) effects [15, 25], and extreme-ultraviolet (XUV) high harmonic generation (HHG) [27-29]." Ref. [29] shows that HHG is not the cause of the XUV emission in these works, but that the emission is rather caused by incoherent fluorescence (see also Sivilis et al., Phys. Rev. Lett. (2013) regarding Ref. [28], and a Comment on Ref. [27]). I would suggest to replace "extreme-ultraviolet (XUV) high harmonic generation (HHG)" with something like "strong-field-ionization induced XUV fluorescence".

Authors: We thank the referee for pointing out our inaccurate phrasing. We have corrected it using language similar to that suggested.

Changes to the Manuscript:

The phrase "extreme-ultraviolet (XUV) high harmonic generation (HHG)" has been replaced with "XUV fluorescence induced by strong-field ionization " in lines 38-39.

Reviewer #2: 7) The authors may want to also refer to early works on field emission in microwave fields (Charbonnier et al., Proc. IEEE (1963) or Furse, J. Vac. Sci. Techn. B (1995)).

Authors: We thank the referee for the suggestion and refer to these works in line 51 in the Introduction.

Changes to the Manuscript:

We have added references [34] and [35] to the revised manuscript. The other references have been renumbered as required.

Response to Reviewer #3:

Reviewer #3: "The paper by Li and Jones is a beautiful and careful study of field-enhanced electron emission from tungsten nano-tips in a THz field. This subject has been an area of considerable interest both from a fundamental and applied perspective. Recent reports have studied the emission using pulses that varied from near-visible to long-wavelength (10 micron). In the current paper, wavelengths (200 microns) that are 10-times longer than previous reports are employed. The results are spectacular, electrons are emitted in excess of 5 keV energy even though the photon energy is less than a millivolt. These observed energies are order of magnitude(s) higher than all previous reports. The reason appears to be linked to the longer wavelength which results in a near-static field as the electron traverses the enhancement region. Enhancement values of 3000 are reported. The study systematically investigates the behavior as a function of field strength and nano-tip size. A simple classical model in an exponentially decaying field nicely reproduces the results. This is a very nice piece of work which will have broad interest in the photonics and strong-field community. I strongly recommend publication in Nature Comm.

Authors: We very much appreciate the positive comments and strong support for publication.

Reviewer #3: "One minor typo on pg 2, line 41: ponderomotive is misspelled"

Authors: Thank you. The misspelling has been corrected in line 37 of the revised manuscript.

Additional Required Formatting Changes:

1. The length of the abstract has been reduced to < 150 words as required.
2. A brief title has been added to all figure legends.
3. Data points and curves in Figures 1 and 3 have been recolored to eliminate red and green, avoiding the need to distinguish between them.

REVIEWERS' COMMENTS:

Reviewer #1 (Remarks to the Author):

The authors responded to all criticism in great detail
and the resulting manuscript is of high quality and can be published as is.

Reviewer #2 (Remarks to the Author):

I have carefully read the revised manuscript and the authors' responses to all reviewer comments. The authors have carefully addressed all comments and have revised the manuscript accordingly. I find this very interesting work clearly suitable for publication.